# Frailty-Focused Movement Monitoring: A Single-Camera System Using Joint Angles for Assessing Chair-Based Exercise Quality

**DOI:** 10.3390/s25133907

**Published:** 2025-06-23

**Authors:** Teng Qi, Miyuki Iwamoto, Dongeun Choi, Noriyuki Kida, Noriaki Kuwahara

**Affiliations:** 1Doctoral Program of Advanced Fibro-Science, Kyoto Institute of Technology, Kyoto 606-8585, Japan; d3851001@edu.kit.ac.jp; 2Department of Social System Studies, Doshisha Women’s College of Liberal Arts, Kyoto 610-0395, Japan; m-iwamoto@dwc.doshisha.ac.jp; 3Faculty of Fiber Science and Engineering, Kyoto Institute of Technology, Kyoto 606-8585, Japan; choide03@kit.ac.jp; 4Faculty of Arts and Sciences, Kyoto Institute of Technology, Kyoto 606-8585, Japan; kida@kit.ac.jp; 5Faculty of Information and Human Sciences, Kyoto Institute of Technology, Kyoto 606-8585, Japan

**Keywords:** frailty, chair-based exercise, older adults, movement correctness, sEMG, machine learning classification, exercise quality monitoring

## Abstract

Ensuring that older adults perform chair-based exercises (CBEs) correctly is essential for improving physical outcomes and reducing the risk of injury, particularly in home and community rehabilitation settings. However, evaluating the correctness of movements accurately and objectively outside clinical environments remains challenging. In this study, camera-based methods have been used to evaluate practical exercise quality. A single-camera system utilizing MediaPipe pose estimation was used to capture joint angle data as twenty older adults performed eight CBEs. Simultaneously, surface electromyography (sEMG) recorded muscle activity. Participants were guided to perform both proper and commonly observed incorrect forms of each movement. Statistical analyses compared joint angles and sEMG signals, and a support vector machine (SVM) was trained to classify movement correctness. The analysis showed that correct executions consistently produced distinct joint angle patterns and significantly higher sEMG activity than incorrect ones (*p* < 0.001). After modifying the selection of joint angle features for Movement 5 (M5), the classification accuracy improved to 96.26%. Including M5, the average classification accuracy across all eight exercises reached 97.77%, demonstrating the overall robustness and consistency of the proposed approach. In contrast, high variability across individuals made sEMG less reliable as a standalone indicator of correctness. The strong classification performance based on joint angles highlights the potential of this approach for real-world applications. While sEMG signals confirmed the physiological differences between correct and incorrect executions, their individual variability limits their generalizability as a sole criterion. Joint angle data derived from a simple single-camera setup can effectively distinguish movement quality in older adults, offering a low-cost, user-friendly solution for real-time feedback in home and community settings. This approach may help support independent exercise and reduce reliance on professional supervision.

## 1. Introduction

Population aging has become a global trend. According to United Nations projections, by 2050, individuals aged 65 years and older will constitute approximately 16% of the global population, a marked increase from 9% in 2019 [1]. The rapid growth of the older adult population has turned age-related health concerns into critical public health challenges. Among these, frailty arises from a decline in multiple physiological systems, leading to increased vulnerability. Commonly observed in older individuals, frailty is strongly associated with adverse outcomes such as falls, disability, hospitalization, and mortality [2]. Research suggests that sustained physical activity can delay the onset and progression of frailty and improve the quality of life among older adults [2,3,4]. Among various physical interventions for older adults, chair-based exercise (CBE) training has garnered considerable attention owing to its safety, simplicity, and suitability for frail individuals [5]. CBE specifically targets those with limited mobility, and studies have shown it can significantly improve multiple functional parameters (e.g., balance, gait speed, and muscle strength), serving as a progressive approach to engage sedentary or frail older adults in exercise [6]. However, to maximize exercise benefits, it is essential to ensure correct movement patterns. Incorrect postures may diminish training effectiveness or even lead to injuries. Consequently, objectively monitoring the correctness of CBE postures performed by older adults in home or community settings has become a key technical challenge in geriatric rehabilitation and exercise-based health interventions.

Recently, advances in sensing and artificial intelligence have prompted the development of numerous systems for posture recognition and action classification, aiming to provide personalized guidance and feedback to older individuals [7,8,9]. In particular, multimodal information fusion has attracted considerable attention. Many studies combine video-based pose estimation with wearable sensors—chief among them surface electromyography (sEMG)—to leverage complementary data sources in recognizing and classifying human actions [10,11]. Such approaches capture skeletal keypoints or pose sequences via standard cameras while simultaneously recording muscle activation through sEMG, thereby enhancing both recognition accuracy and quality assessment [10]. However, these systems frequently require high-dimensional data acquisition and sophisticated hardware or algorithms, limiting their feasibility in typical communities and home settings. This challenge is especially pronounced among older adults with reduced cognitive or operational capacities.

Previous work with a focus on practical deployment introduced a lightweight approach for assessing movement correctness during CBE training [12]. Relying solely on a monocular camera and an open-source human pose estimation framework (MediaPipe), the method extracts eight key upper-limb and trunk joint coordinates, from which kinematic parameters—particularly joint angles—are derived to determine posture accuracy. This approach may achieve better acceptance among older users by eliminating additional wearable sensors. MediaPipe, developed by Google, offers real-time, multi-person body pose-estimation models using off-the-shelf cameras in smartphones or computers [13]. Additionally, compared to solutions that rely on depth cameras, MediaPipe is both inexpensive and easily deployable, making it well-suited for home-based self-training in older populations [13]. Studies have validated its reliability in rehabilitation contexts, demonstrating that MediaPipe’s measurements of joint range of motion are largely consistent with clinical goniometric data and correlate strongly with laboratory 3D motion capture systems [14,15].

Nonetheless, image-based pose-estimation algorithms (MediaPipe) generally provide two-dimensional coordinates susceptible to frame-by-frame fluctuations (jitter), occlusions, and lighting variations [16,17,18]. Such instabilities are especially pronounced in static or slow movements, often complicating posture evaluation. To enhance both robustness and temporal consistency, the present study proposes using joint angles rather than raw coordinate points for action analysis. Because joint angles introduce structural constraints by relating multiple keypoints, local jitter errors can be partially mitigated, increasing the stability of the extracted pose features.

Previous methods, compared with conventional multimodal algorithms, substantially reduce input dimensionality and system complexity, enabling more practical, real-time monitoring for older adults in community settings [12]. Previous work leveraged MediaPipe-derived joint-angle changes to autonomously distinguish correct and incorrect CBE forms, reaching an accuracy of over 97% [12]. This underscores the feasibility of evaluating CBE movement quality through vision-based pose analysis [12].

This study aims to validate and expand the efficacy of using low-dimensional features by integrating sEMG and related data based on this framework. This study pursues two principal objectives. First, it investigates the influence of correct versus incorrect CBE movements on muscle loading among older adults via multi-channel sEMG recordings. The authors hypothesize that properly executed movements yield stronger and more appropriate muscle activation, whereas incorrect postures may compromise muscular effort. In order to test this hypothesis, three analytical approaches are used: (1) plotting sEMG waveforms to observe the temporal patterns of muscle activation under correct versus incorrect conditions, anticipating similar overall movement trajectories but pronounced amplitude differences; (2) extracting the RMS (root mean square) peak values from the sEMG signals and statistically comparing boxplot distributions to assess relative muscle output intensities; (3) generating scatter plots of sEMG features across participants to investigate individual variations as well as overall trends. This analysis provides objective evidence supporting the importance of correct form in exercise by quantifying the differences in muscle effort.

Second, this study evaluates whether joint angles alone can accurately and correctly distinguish incorrect movements. Although multimodal sensor fusion can improve classification performance, relying solely on joint-angle data extracted from a single camera would simplify system deployment and reduce the user burden among older adults. Hence, this research compares the differences in both joint-angle data and sEMG signals for correct and incorrect exercises.

In summary, the study focuses on chair-based exercise for older adults by combining computer vision-based pose evaluation with sEMG analysis to elucidate the relationship between movement accuracy and muscular load, as well as to explore the viability of automatic exercise quality classification under minimal input conditions. These findings have direct implications for the development of user-friendly and intelligent rehabilitation tools for older adults and offer valuable baseline data and methodological guidance for future investigations in this area.

To be precise, the system is a camera-only solution operable without any wearable sensor. The present inclusion of sEMG aims solely to validate that monocular camera-derived joint angles reliably correlate with underlying muscle activation.

## 2. Materials and Methods

### 2.1. Participants

In this study, a surface EMG sensor was employed only during the experimental phase to benchmark camera-based kinematic measurements. The long-term goal is to deploy a system requiring only an RGB camera and no wearable device.

Following informed consent, older adult participants were fitted with surface electromyography (sEMG) equipment.

A total of 20 community-dwelling older adults were recruited for this study (10 males and 10 females; mean age: 72.35 years; males’ mean age: 73 years; females’ mean age: 71.7 years). Due to ethical and procedural considerations, participants’ height, weight, and BMI were not measured. Inclusion criteria were limited to age (above 65 years) and the ability to independently perform basic daily activities. No specific exclusion criteria were applied regarding prior musculoskeletal conditions or history of injury to improve the generalizability and practical applicability of the study findings. Participants were not required to have previous experience with resistance training. Informal interviews conducted during the experiment revealed that most participants regularly engaged in daily walking, but none participated in structured muscle training programs.

### 2.2. Target Exercises

A redesigned training chair, developed in previous work based on the principles of chair-based exercise (CBE) and ergonomic considerations, has demonstrated improved stability for elderly users [12]. This design was particularly effective in promoting correct posture execution during sit-to-stand resistance training. However, for the remaining eight CBE movements, reliance on the chair alone was insufficient to ensure correct execution, thereby highlighting the need for further investigation. Therefore, the present study conducts an in-depth analysis of these eight movements in Table 1 to evaluate motion correctness and muscle activation patterns.

### 2.3. Experimental Procedure

Under the guidance of the research team, each participant performed eight distinct CBE training exercises. For each exercise, participants completed 10 repetitions in correct form and 10 repetitions in an intentionally incorrect form, reflecting common errors observed among the elderly. A video camera positioned at a 45° angle to the right-front side of the participant captured the entire procedure for subsequent kinematic and qualitative analysis.

### 2.4. Criteria for Evaluating Movement Correctness

In this study, movement correctness was assessed using six key joint angles, as illustrated in Figure 1. Each of the eight exercises was performed in both correct and incorrect forms. The correct execution was defined in accordance with standard CBE movement guidelines, while the incorrect execution simulated the common deviations observed among elderly participants [19,20,21,22]. Figure 2, Figure 3, Figure 4, Figure 5, Figure 6, Figure 7, Figure 8 and Figure 9 provide representative examples of both execution types, with red markers clearly highlighting the differences in joint positions and movement trajectories.

For M1, the correct posture requires the lower leg to remain perpendicular to the ground during the exercise. Two common errors observed are placing the foot too far forward or backward, which alters the lower leg angle and compromises the intended posture. For M2, the exercise should be performed with the thigh kept as straight as possible while executing ankle movements. Deviation from thigh alignment may reduce the effectiveness of the exercise. M3 is relatively difficult for older adults. A frequent incorrect form involves performing leg abduction with both feet in contact with the floor. In contrast, the correct form requires raising both legs while performing the abduction movement. In M4, the correct execution involves lifting the thigh while maintaining a consistent knee angle. A common mistake is allowing the knee to hang vertically downward, resulting in unintentional flexion. For M5, participants are instructed to keep the torso upright without leaning forward. The backrest of the chair is intended only to assist in maintaining balance, not to support the body weight during execution. In M6, proper form requires that the knees do not extend beyond the toes during movement. This helps prevent excessive loading on the knee joints. During M7, the correct posture involves raising the thigh laterally while keeping the upper body stable and upright. Tilting or rotating the trunk during the lift is considered an error. For M8, the thigh should be lifted to a horizontal position to ensure adequate training intensity. Insufficient thigh elevation may reduce the load on the targeted muscle group.

### 2.5. sEMG Data Collection

Surface electromyography (sEMG) signals were collected using a wireless dry-type sEMG sensor (Model: LP-WS1221, LOGICAL PRODUCT Corporation, Fukuoka, Japan) at a sampling frequency of 1000 Hz. In total, five muscle sites were selected for measurement across the eight targeted chair-based exercises.

Although the iliopsoas is the primary target muscle in Movement 8 (M8), sEMG is known to have limited sensitivity for deep muscles due to signal attenuation and interference from overlying tissues. While specialized sEMG recording techniques for the iliopsoas have been proposed [23], they often involve invasive or non-standard electrode placements, which are not suitable for use with older adult populations. To ensure both reliability and practicality, the quadriceps femoris—a synergist muscle group engaged in hip flexion—was chosen instead for sEMG recording.

Similarly, in Movement 4 (M4), which targets the iliopsoas and abdominal muscles, the distinguishing factor between correct and incorrect execution lies in the control of knee joint angles. Since quadriceps activation is directly involved in knee extension control, the same measurement site was used for this movement as well [24].

All sEMG electrode placements followed the SENIAM recommendations, and details of the muscle sites and corresponding movements are summarized in Table 2.

### 2.6. sEMG Data Processing and Analysis Workflow

#### 2.6.1. sEMG Data Preprocessing

The authors first applied a 50 Hz notch filter to remove electrical interference from the power grid. A Butterworth bandpass filter (20–450 Hz) was then used to eliminate low-frequency motion artifacts and high-frequency noise. Given that the movements in this experiment were relatively slow (approximately 20 s each) and remained stable throughout each trial, a 100 ms window length with 50% overlap was selected to capture long-term electromechanical activity across the ten repetitions performed by each participant.

#### 2.6.2. sEMG Data Visualization

Because data collection durations varied among participants, the authors normalized the Window_Index variable to the (0,1) interval to ensure a uniform temporal scale for cross-dataset comparisons. The data were subsequently resampled to 500 time points using a linear interpolation method, thereby aligning all the time series to a common reference frame. A Gaussian filter (σ = 10) was then applied to reduce high-frequency fluctuations and enhance the visual clarity of sEMG trends. Finally, the authors plotted the smoothed waveforms for sEMG datasets on a single figure, allowing for direct comparisons of signal variations under identical axes.

#### 2.6.3. sEMG Signal Analysis

For each participant and each movement, the root mean square (RMS) values of sEMG signals were calculated separately for both correct and incorrect executions. The overall average RMS across the entire time series was used to quantify muscle activation intensity and to compare differences between correct and incorrect forms.

In addition, each participant performed 10 repetitions of both correct and incorrect movements. For each trial, the peak RMS value was extracted, and the distribution of these peak values was statistically analyzed. This allowed the authors to evaluate not only the magnitude but also the consistency of muscle activation between conditions. The analysis aimed to determine whether correct executions required greater and more variable muscular effort compared to incorrect forms, potentially indicating greater demands on motor control.

### 2.7. Joint Data Processing and Analysis Workflow

The joint angle processing approach in this study followed the same methodology as described in previous work [12]. Human body keypoints were extracted from the experimental video data using the MediaPipe pose estimation framework, which provides reliable detection of 33 anatomical landmarks in two-dimensional space.

For this study, eight keypoints related to lower-limb kinematics were selected: the left and right shoulders (IDs 11 and 12), hips (23 and 24), knees (25 and 26), and ankles (27 and 28), as shown in Figure 10 [25]. The x-coordinates and y-coordinates of these landmarks were obtained for each video frame. Joint angles were computed using vector-based geometric calculations between relevant segments (e.g., hip–knee–ankle) to quantify motion patterns.

The average joint angles were then calculated across all frames for each repetition. The resulting data were aggregated to compare the mean joint angles of correct versus incorrect movements across all participants, enabling the evaluation of postural and kinematic differences between conditions.

### 2.8. Feature Extraction from Joint Angles

To characterize the dynamic patterns of joint angles over time, a sliding window approach was applied to the time series data. Given that the video recordings had a frame rate of 30 frames per second (fps), a window size of 30 frames (1 s) with a 50% overlap (15 frames) was used to segment the data.

For each window segment, a set of descriptive statistical features was computed to capture the temporal characteristics of joint motion:

Mean: represents the average joint angle value within the window, reflecting the central tendency of joint posture.

Minimum and Maximum: captures the extreme joint positions within the window, useful for identifying peak extension or flexion.

Median: provides a robust measure of central tendency, less sensitive to outliers.

Standard Deviation (SD): measures variability, indicating the consistency or fluctuation of joint movement.

In addition to time-domain features, frequency-domain characteristics were also extracted using power spectral density (PSD) analysis:

Welch’s Method: applied to each window to estimate the power spectrum of joint angle signals.

Spectral Features: within the frequency band of 1–20 Hz (relevant for slow, voluntary human movement), the following features were calculated: mean power, peak power, minimum power, median frequency, and standard deviation of power.

All extracted features were stored for subsequent use in movement classification models, enabling joint angle-based recognition of correct versus incorrect postures.

### 2.9. SVM Model Evaluation

To classify the correctness of movement execution based on the extracted features, a support vector machine (SVM) classifier was employed. The input feature set consisted of both time-domain and frequency-domain statistical parameters derived from joint angle data. The corresponding binary labels (correct vs. incorrect action) were assigned to each sample based on the experimental design.

Given the relatively small sample size and the need for participant-level generalization, leave-one-out cross-validation (LOOCV) was adopted as the evaluation strategy. In this approach, the model is trained on all but one sample, which is then used for testing, and the process is repeated for each sample. This method ensures unbiased performance estimation, especially in limited data scenarios.

To address potential class imbalance between correct and incorrect samples, class weights were automatically computed and incorporated into the model training. Classification performance was quantified using average accuracy across all LOOCV iterations, along with standard classification metrics such as precision, recall, and F1-score.

While the present study includes an analysis of some movements also examined in our previous publication [12], it is based on a different dataset. Furthermore, the analytical methods for specific movements have been modified with the primary goal of enhancing classification performance. Importantly, the focus of this study differs from our previous work, shifting toward a more detailed investigation of sEMG signal characteristics and the classification performance based solely on joint angles.

## 3. Results

### 3.1. Analysis of sEMG Waveforms

To qualitatively examine muscle activation patterns, sEMG waveforms were visualized for a representative participant randomly selected from the 20 elderly subjects. The sEMG signals correspond to both correct and incorrect executions of a single CBE movement.

As illustrated in Figure 11, the waveform corresponding to the correct execution exhibited a consistently higher amplitude than that of the incorrect form, indicating a greater level of muscle activation. Despite differences in signal magnitude, the overall temporal patterns and shape of the waveforms were similar across both conditions, suggesting that the movement sequence was preserved while the exertion level varied.

### 3.2. Peak RMS Distribution Analysis

To further investigate differences in muscle activation between correct and incorrect executions, peak RMS values of sEMG signals were extracted from each of the 10 repetitions per condition across all eight CBE movements. Boxplots were generated for a representative elderly participant randomly selected from the study cohort.

As shown in Figure 12, the correct execution trials generally exhibited higher peak RMS values compared to incorrect executions across most movements. In addition, the distribution of peak values during correct executions was broader, indicating greater variability in muscular effort. In contrast, the incorrect execution group displayed more clustered peak values with narrower interquartile ranges, suggesting more consistent—yet overall lower—muscle activation.

### 3.3. Scatterplot Analysis

To explore the inter-individual variation in muscle activation between correct and incorrect executions, scatterplots were constructed for each of the eight chair-based exercises. In each plot, every data point represents one participant, where the x-axis indicates the mean sEMG value of the correct executions and the y-axis indicates the mean sEMG value of the incorrect executions.

As shown in Figure 13 (Action 8 example), the majority of data points lie below the identity line (red dashed line), indicating that for most participants, the mean muscle activation during correct execution was higher than during incorrect execution. Furthermore, for each of the eight CBE exercises, paired *t*-tests were conducted to compare the mean sEMG values between correct and incorrect executions. The results revealed statistically significant differences across all movements (*p* < 0.001), thereby confirming that correct posture consistently elicited stronger muscle activation.

While at first glance the asymmetric distribution of data points—mostly concentrated below the identity line—may appear unexpected, it is in fact a meaningful representation of this consistent pattern. It reflects that correct posture systematically leads to stronger muscle engagement. Some inter-individual variation exists, with a few points near or above the line, but the overall visual and statistical evidence reinforces the conclusion. 

### 3.4. Summary of Joint Angle and sEMG Differences

Table 3 presents the mean and standard deviation (SD) of joint angles and sEMG amplitudes for correct and incorrect executions across eight chair-based exercises. Each value represents the average across 20 participants.

In most actions, joint angle values between correct and incorrect executions showed clear differences. For example, in M1-A, the average joint angle in the correct group was 177.99° ± 0.93°, compared to 172.10° ± 2.15° in the incorrect group. Similar separations were observed in other actions. The SDs of the joint angles were generally low across movements.

For sEMG data, correct executions showed consistently higher mean values than incorrect executions. In M2, the correct condition recorded a mean sEMG value of 0.11 ± 0.06 mV, while the incorrect condition recorded 0.06 ± 0.05 mV. However, the sEMG results exhibited higher variability, with larger standard deviations observed across nearly all movements.

### 3.5. Enhancement of SVM-Based Classification for M5

A support vector machine (SVM) was constructed to classify correct versus incorrect executions of M5 using the knee–ankle angle as the primary input feature. This angle was selected due to its stable distribution across execution types in the dataset. Compared to the previous study, which used the shoulder–hip angle, the use of the knee–ankle angle resulted in improved classification performance [12]. With a balanced dataset for correct and incorrect samples, the model achieved an average classification accuracy of 96.26% in cross-validation and 97.48% on an independent test set, as shown in Table 4, exceeding the previously reported accuracy of 95.51%.

In Table 4, the support column indicates the number of samples for each class used in the evaluation. It reflects the distribution of the dataset and helps interpret the precision, recall, and F1-score values by showing how many instances contributed to each metric. This information is important to assess the reliability of the classification performance, especially for classes with fewer samples.

## 4. Discussion

Based on the experimental results, it was confirmed that using joint angles alone allows for the accurate evaluation of movement correctness. At the same time, the limitations of muscle activation indicators, such as surface electromyography (sEMG) in chair-based exercise (CBE) training for older adults, were also revealed. In most tested actions, the classification based on joint angle data captured by a single camera showed clear separability between correct and incorrect executions. Although the sEMG data also indicated higher muscle activation during correct movements, it was found that due to significant inter-individual variability among older adults, sEMG is not suitable as a standalone indicator for judging movement correctness. Moreover, using sEMG requires measuring each individual’s maximum voluntary contraction (MVC), which is difficult to implement under practical training conditions.

The experimental data showed that across all eight CBE exercises, correct and incorrect movements generally presented stable and distinguishable ranges of joint angles. For instance, in certain actions, the average joint angle during correct execution reached approximately 178°, while in incorrect execution it was only around 172°, with relatively small standard deviations in both cases. This robust difference demonstrates that when older participants fail to perform a movement correctly, the deviation in joint range and posture is observable and quantifiable.

From an application perspective, such joint angle data can be obtained using a single camera combined with a lightweight pose estimation algorithm such as MediaPipe. Even with a simple classification model, an accuracy rate exceeding 96% was achieved. These results support the feasibility and precision of using joint angles to evaluate movement quality and indicate the potential for deploying such systems in home or community rehabilitation settings. As the method relies solely on visual input, it significantly reduces hardware complexity and user burden, making it more acceptable for older adults and enhancing adherence. This also continues the concept proposed in previous work [12] regarding low-dimensional, practical monitoring frameworks.

In addition, sEMG measurements further validated that correct posture generally results in greater muscle activation, emphasizing the necessity of maintaining proper form during CBE training. However, considerable inter-individual differences in absolute sEMG values and variation ranges were observed. Some older adults with lower muscular strength exhibited lower sEMG peaks during “correct” execution than those of other participants did during “incorrect” execution. Due to the inherent physiological diversity in older populations—such as differences in muscle mass and functional capacity—relying solely on sEMG thresholds or average values is not broadly applicable. Person-specific calibration or adaptive adjustments would be required, yet such individual calibration is often impractical in real-world applications.

Moreover, sEMG technology itself has inherent limitations. As it primarily records surface muscle activity, it does not accurately reflect the involvement of deeper or stabilizing muscles (e.g., the transversus abdominis or iliopsoas). The signals are also susceptible to artifacts caused by electrode adhesion, skin impedance, and other external factors, leading to increased noise and larger standard deviations. In non-clinical environments such as homes or community centers, the absence of professional operation and robust signal processing further hinders the precision of sEMG-based movement assessment. Therefore, while sEMG can offer physiological insight, its standalone use in evaluating movement correctness—particularly in older adults—remains limited in practicality.

Furthermore, due to variability in the physical condition of the older participants recruited in this study, the “correct” executions may have still contained minor deviations in some individuals, especially those with limited joint flexibility or motor control. Similarly, the level of “incorrect” executions also varied across individuals.

Previous validation studies have compared inertial motion capture systems such as Xsens MVN with marker-based optical systems like VICON [26,27]. However, it is important to note that these systems typically rely on strict frontal, sagittal, or transverse plane views, whereas our system uses a single monocular camera positioned at a 45-degree angle, simulating a more realistic and flexible setup for home-based implementation [26,27]. While this orientation improves practicality and ease of use, it may affect the depth and rotational accuracy of some movements, especially compared to 3D marker-based systems. It is important to note that these commercial systems require multiple cameras or body-worn sensors, controlled environments, specialized software, and trained operators. Furthermore, their high equipment and installation costs make them impractical for home or community-based rehabilitation.

In contrast, the system proposed in this study utilizes only a monocular camera and a lightweight pose estimation algorithm, which provides a more accessible, low-cost solution. Future research should include quantitative comparisons between our method and these commercial systems to validate its accuracy and robustness under varying real-world conditions.

## 5. Limitation

This study involved only 20 older adult participants (10 males and 10 females). While the results showed consistent trends, this relatively small sample size limits the generalizability of the findings. In addition, physical metrics such as height, weight, and BMI were not recorded, which may introduce uncontrolled variability in joint kinematics and muscle activity. Therefore, this study should be considered a preliminary validation. Larger-scale experiments involving more demographically diverse participants and comprehensive physical profiling will be essential to further assess the robustness and applicability of the proposed system.

The experiments were conducted in controlled laboratory settings with adequate lighting and minimal background interference. However, the system has not yet been tested in real-world home environments, where lighting conditions, background clutter, and camera placement may vary significantly. These factors could affect the robustness and reliability of MediaPipe-based pose estimation and should be explored in future field studies.

This study did not compare the proposed system with commercial systems such as VICON or Xsens MVN. Such comparisons would help assess the trade-offs between accessibility and accuracy and are planned for future work.

## 6. Conclusions

This study demonstrated a lightweight movement quality monitoring method based on a monocular camera and pose estimation algorithm, which effectively distinguishes between correct and incorrect executions of chair-based exercises in older adults. These outcomes were consistent with the trends observed in sEMG measurements, where correct executions generally elicited higher levels of muscle activation (*p* < 0.001), indicating the physiological plausibility of the proposed method. In this study, only the classification model for Movement 5 was reconstructed using the knee–ankle joint angle as the primary feature, which demonstrated superior discriminatory stability compared to previously used features. After modifying the selection of joint angle features for Movement 5 (M5), the classification accuracy improved to 96.26%. The classification models for the remaining CBE actions were inherited from prior work [12], as their performance had already met the required accuracy. Moreover, the low cost and portability of a single-camera system support its deployment in home and community environments, offering potential for self-managed rehabilitation or exercise programs among older adults.

This study also revealed notable challenges associated with using sEMG as a stand-alone indicator of CBE movement correctness, primarily due to high inter-individual variability. Without sufficient personalization or calibration, its applicability as a universal quantitative metric remains limited. In contrast, joint angles provide a more uniform and stable standard for assessment, directly reflecting movement posture. Therefore, from a practical perspective, joint angle-based monitoring systems are more feasible and better aligned with older adults’ expectations for ease of use and operational simplicity.

While previous studies have widely confirmed that proper execution of sit-to-stand training significantly enhances lower-limb strength and function, little attention has been given to whether other types of CBEs exhibit similar differences in muscle activation or kinematic profiles between correct and incorrect performances. This study fills that gap by providing the first systematic physiological comparison—via both sEMG and joint angle analysis—of eight common CBE movements, thus offering a theoretical foundation for refining exercise guidelines and developing personalized rehabilitation strategies.

Future research will aim to incorporate real-time posture monitoring and feedback mechanisms, enabling users to receive immediate correction cues during exercise and thereby replicating the benefits of in-person supervision in home environments. Additionally, the study will be expanded to include a larger and more diverse participant pool and a broader range of exercise types to assess the generalizability and robustness of the proposed method. Stratified evaluations based on age groups or functional levels (e.g., individuals with mild to moderate impairments) will also be considered for more precise adaptation and application.

Although the system reduces hardware complexity by using only a monocular camera, it still requires a technical setup to run the software. This may present usability challenges, particularly for older adults operating the system independently without technical support. Future development should focus on improving user interface simplicity and automating calibration to enable home use without professional assistance.

## Figures and Tables

**Figure 1 sensors-25-03907-f001:**
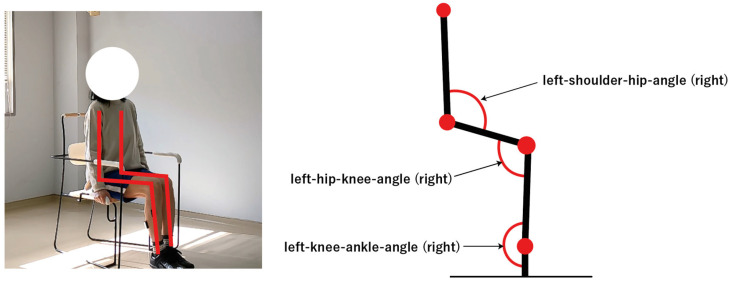
Figure 1 shows the positions of six key joint angles derived from video-based pose estimation.

**Figure 2 sensors-25-03907-f002:**
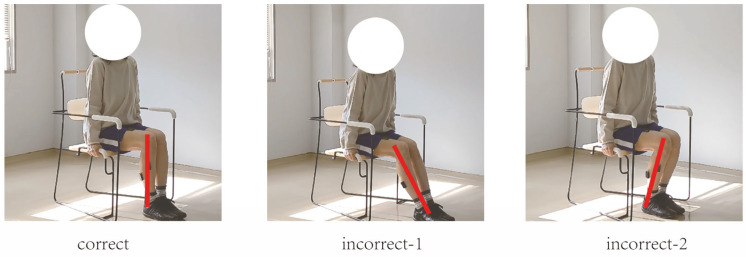
Figure 2 shows visual examples of the correct and incorrect actions for M1.

**Figure 3 sensors-25-03907-f003:**
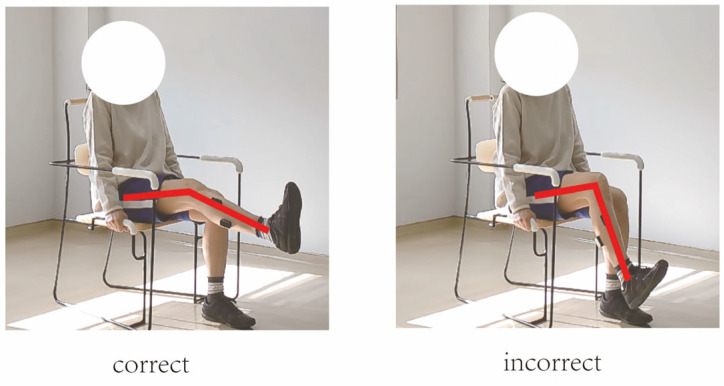
Figure 3 shows visual examples of the correct and incorrect actions for M2.

**Figure 4 sensors-25-03907-f004:**
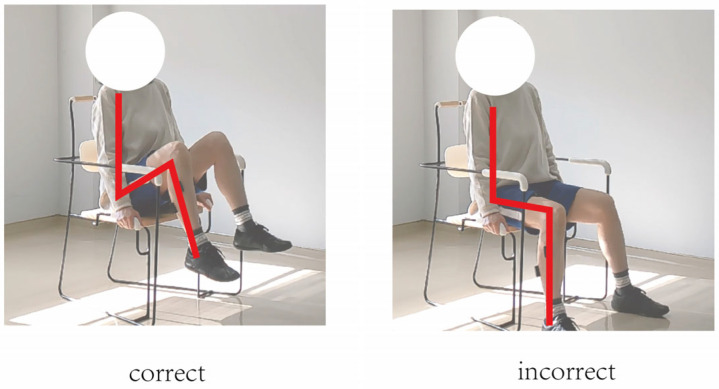
Figure 4 shows visual examples of the correct and incorrect actions for M3.

**Figure 5 sensors-25-03907-f005:**
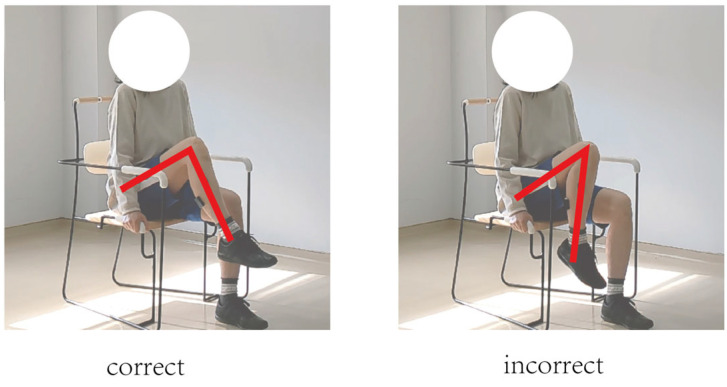
Figure 5 shows visual examples of the correct and incorrect actions for M4.

**Figure 6 sensors-25-03907-f006:**
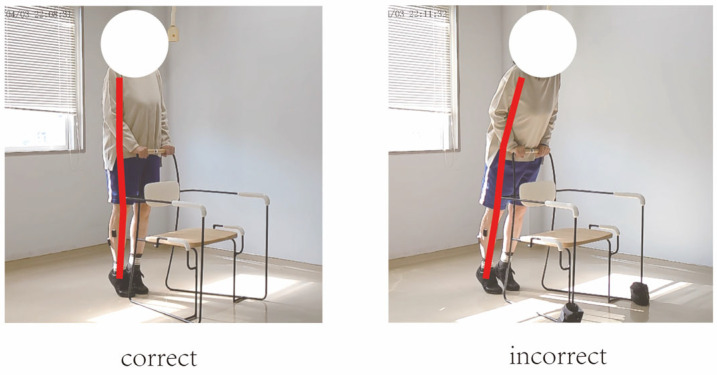
Figure 6 shows visual examples of the correct and incorrect actions for M5.

**Figure 7 sensors-25-03907-f007:**
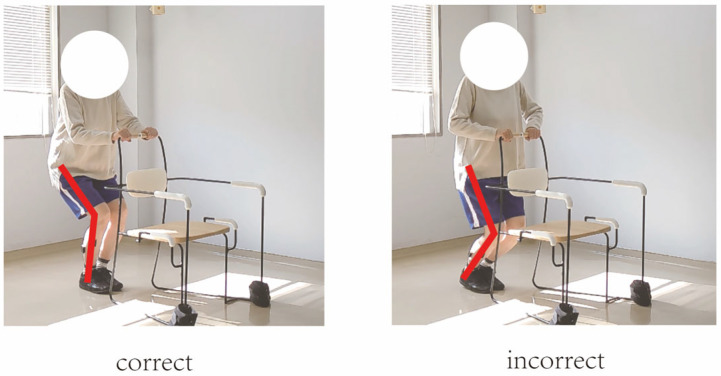
Figure 7 shows visual examples of the correct and incorrect actions for M6.

**Figure 8 sensors-25-03907-f008:**
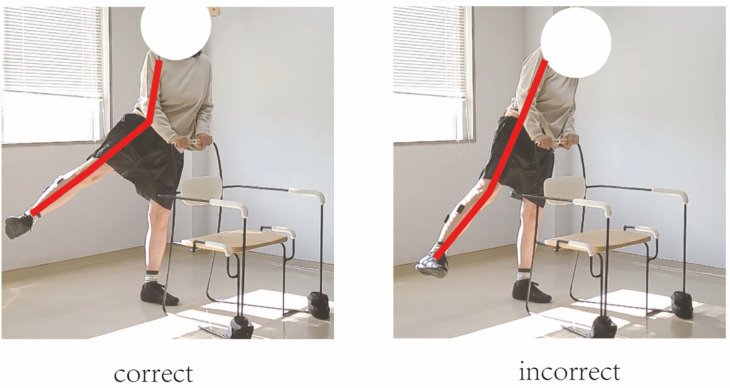
Figure 8 shows visual examples of the correct and incorrect actions for M7.

**Figure 9 sensors-25-03907-f009:**
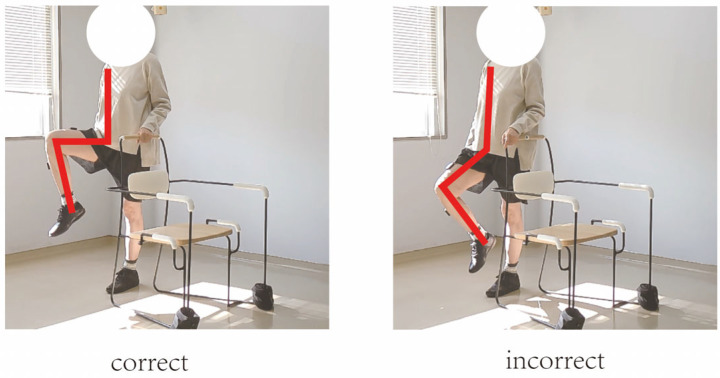
Figure 9 shows visual examples of the correct and incorrect actions for M8.

**Figure 10 sensors-25-03907-f010:**
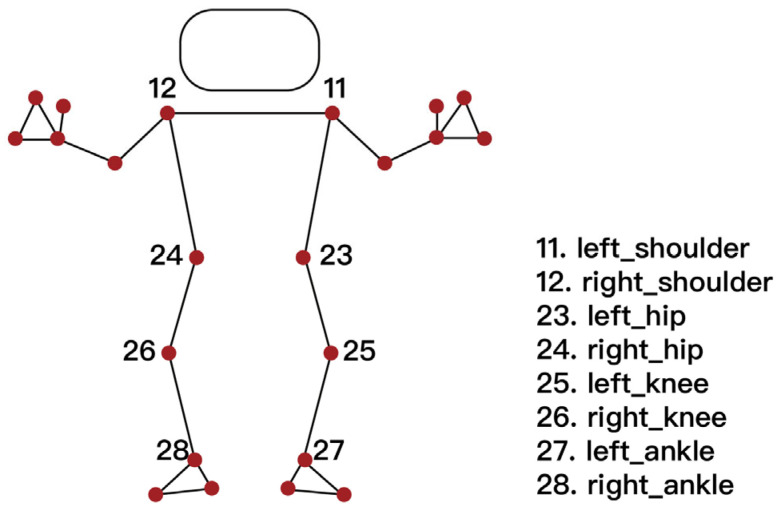
Figure 10 shows pose landmarks.

**Figure 11 sensors-25-03907-f011:**
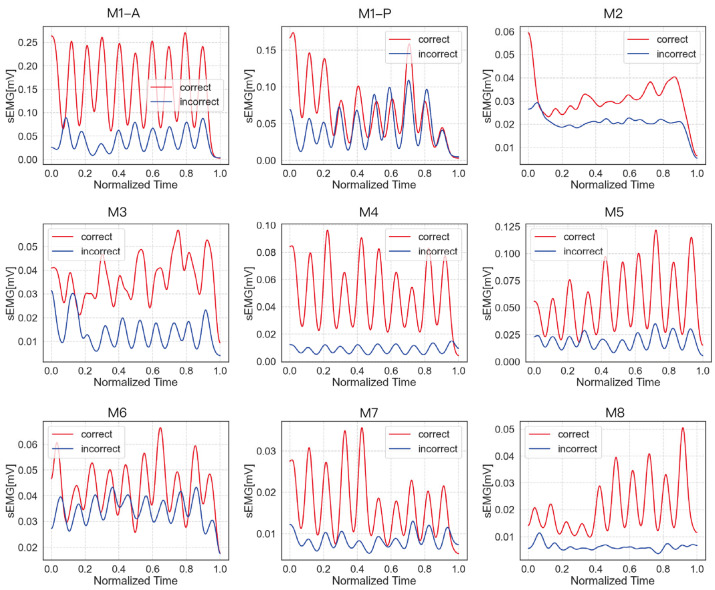
Figure 11 shows the waveform plots of each action under both correct and incorrect execution conditions. The x-axis represents the normalized time, while the y-axis indicates the sEMG signal amplitude. During M1, dorsiflexion and plantarflexion engage different muscle groups. Therefore, separate sEMG measurements were conducted. M1-A corresponds to the measurement of the anterior tibialis, while M1-P corresponds to the measurement of the peroneal muscles. Notably, different incorrect execution types were used for each measurement: the anterior tibialis was assessed under incorrect 1 of M1, whereas the peroneal muscles were assessed under incorrect 2. The correct execution form was identical for both measurements.

**Figure 12 sensors-25-03907-f012:**
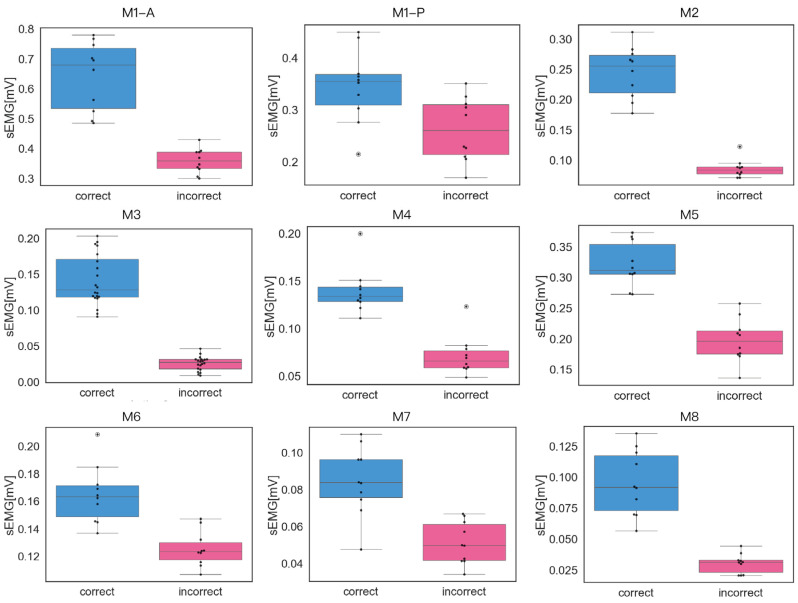
Figure 12 shows boxplot comparisons of sEMG peak values for one older participant, showing ten correct and ten incorrect executions for each movement. During M1, dorsiflexion and plantarflexion engage different muscle groups. Therefore, separate sEMG measurements were conducted. M1-A corresponds to the measurement of the anterior tibialis, while M1-P corresponds to the measurement of the peroneal muscles. Notably, different incorrect execution types were used for each measurement: the anterior tibialis was assessed under incorrect 1 of M1, whereas the peroneal muscles were assessed under incorrect 2. The correct execution form was identical for both measurements.

**Figure 13 sensors-25-03907-f013:**
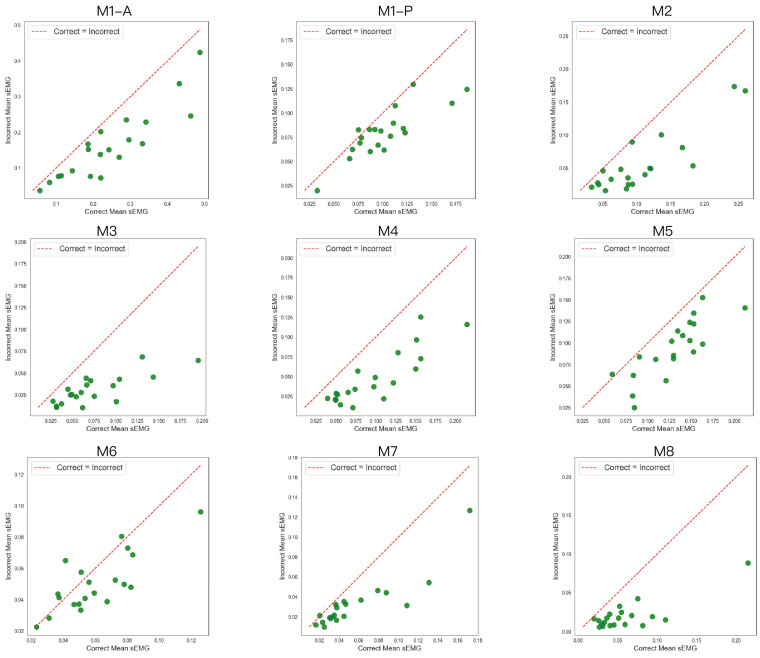
**Figure 13** shows the average sEMG values during correct and incorrect executions for each movement across all 20 participants. The green dots in the figure represent the participants. During M1, dorsiflexion and plantarflexion engage different muscle groups. Therefore, separate sEMG measurements were conducted. M1-A corresponds to the measurement of the anterior tibialis, while M1-P corresponds to the measurement of the peroneal muscles. Notably, different incorrect execution types were used for each measurement: the anterior tibialis was assessed under incorrect 1 of M1, whereas the peroneal muscles were assessed under incorrect 2. The correct execution form was identical for both measurements.

**Table 1 sensors-25-03907-t001:** A summary of the movements examined in this study is presented in Table 1.

Movement	Muscle Area(s)	Key Action Description
M1	Anterior tibialis and peroneal muscles	Performed sitting or standing, alternately lifting the toes and then the heels to strengthen the muscles on the front and outer sides of the foot.
M2	Quadriceps	Lift one leg while seated, extending the knee. Then, slowly move the leg back and forth to stimulate the quadriceps.
M3	Tensor fasciae latae, sartorius, and gluteus medius	Start seated with knees together, then open and close the legs laterally.
M4	Iliopsoas, quadriceps, and abdominal muscles	Lift the knees towards the chest while seated to strengthen the iliopsoas and abdominal muscles.
M5	Peroneal muscles	While standing, raise and lower the heels to strengthen the peroneal and other lower leg muscles.
M6	Quadriceps	While standing, strengthen the quadriceps by raising the heels while bending the knees.
M7	Gluteus medius	While standing, raise one leg to the side to train the gluteus medius and improve balance.
M8	Iliopsoas	Perform high knee lifts to train the iliopsoas.

**Table 2 sensors-25-03907-t002:** Table 2 shows the muscle groups measured for each movement.

Movement	Muscle Group	Electrode Placement Description
M1	Anterior tibialis	Lies on the front of the lower leg, just lateral to the tibia.
M1, M5	Soleus muscles	Lies deep in the calf, midway between the fibular head and lateral malleolus on the lateral side.
M2, M4, M6, M8	Quadriceps	Lies on the front of the thigh and consists of four muscles that extend the knee.
M3	Tensor fasciae latae	2 cm below the anterior superior iliac spine, angled slightly laterally.
M7	Gluteus medius	On the lateral aspect of the pelvis, one-third of the distance from the iliac crest to the greater trochanter.

**Table 3 sensors-25-03907-t003:** Table 3 shows the overall average joint angles and average sEMG values across all 20 participants.

Moment	Angle (°)		sEMG(mV)	
	Mean ± SD		Mean	
	Correct	Incorrect	Correct	Incorrect
M1-A	177.99 ± 0.93	172.10 ± 2.15	0.24 ± 0.12	0.16 ± 0.10
M1-P	178.02 ± 0.75	173.27 ± 1.84	0.10 ± 0.04	0.08 ± 0.03
M2	148.53 ± 17.63	125.84 ± 17.48	0.11 ± 0.06	0.06 ± 0.05
M3	107.52 ± 14.31	131.30 ± 12.26	0.08 ± 0.04	0.03 ± 0.02
M4	171.21 ± 3.43	176.38 ± 1.71	0.10 ± 0.05	0.05 ± 0.03
M5	176.95 ± 1.10	173.19 ± 1.58	0.13 ± 0.04	0.09 ± 0.03
M6	175.49 ± 1.75	170.74 ± 2.05	0.06 ± 0.02	0.05 ± 0.02
M7	170.97 ± 3.41	175.56 ± 1.86	0.06 ± 0.04	0.03 ± 0.03
M8	141.78 ± 5.39	166.10 ± 2.99	0.06 ± 0.04	0.02 ± 0.02

**Table 4 sensors-25-03907-t004:** Table 4 shows the performance metrics and relevant details of the support vector machine (SVM) model retrained specifically for M5.

Average Accuracy M5: 96.26%				
	Precision	Recall	f1-Score	Support
0	96%	96%	96%	451
1	97%	96%	97%	591
accuracy			96%	1042
macro avg	96%	96%	96%	1042
weighted avg	96%	96%	96%	1042

## Data Availability

The authors will make the raw data supporting this article’s conclusions available upon request.

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
