# Peer review of "Frailty-Focused Movement Monitoring: A Single-Camera System Using Joint Angles for Assessing Chair-Based Exercise Quality"

_sensors, 2025, doi:10.3390/s25133907_

Round 1
Reviewer 1 Report
Comments and Suggestions for Authors
This is a interesting study with clear practical application but there are places where it is not clear that your value of 96% accuracy of the SVM relates to a single exercise which is standing. Why was this chosen?
What is the accuracy for the other exercises, would it not be relevant to also assess this for a seated exercise? This would then cover the main positions in your exercise series and provide more information on the value of the method for real world use, where seated exercises are likely to be incorporated into supportive exercise plans.
This needs to be more clear in your abstract and also in your discussion so that the reader understands what that accuracy relates to when reading those sections as it not explicitly stated.
Minor comment: L21- data to two decimal places is sufficient and the same for L463
Reviewer 2 Report
Comments and Suggestions for Authors
The article is devoted to the development of a simple method for assessing the quality of performing eight types of exercises in a chair based on the analysis of joint angles obtained using a single camera and a MediaPipe model. Additionally, the authors analyze the sEMG data of the main muscles with correct and incorrect exercise technique. The paper also solves the classification problem using the SVM model. Differences between correct and incorrect exercise performance were revealed (p < 0.001) in both kinematics and muscle activity. The results obtained make the work relevant, and I also liked the fact that the authors did not just choose random exercises, but focused on specific muscle groups in order to cover the maximum variety. The authors clearly described the control group. The list of sources is up-to-date. Thus, the level of the article and its scientific novelty are very worthy.
I had a few questions after reading it:
1) I didn't really understand Figure 13. As I initially thought, there would be dots on both sides of the correct-incorrect line, but basically all the actions have dots concentrated on one side. I ask the authors to explain this drawing in more detail.
2) I also did not understand why the authors developed a model for classifying the correctness of actions only for the M5 exercise, and not for all at once. The experiment was performed in laboratory conditions with lighting and background adequate for the MediaPipe test. A field check (home environment, interference, different cameras) has not been carried out, which makes it difficult to assess the stability of the system in real practice.
3) Perhaps it is worth paying more attention to the efficiency of the method in conditions of poor lighting, to discuss the implementation of the approach in real practice and conditions, including a complex background.
Thus, I recommend the article for publication after explaining/clarifying the listed issues, which relate to minor improvements.
Reviewer 3 Report
Comments and Suggestions for Authors
The authors presented a work about the kinematic assessment of the quality of simple exercises for elderly people's rehabilitation and well-being using a monocular camera and sEMG signals. My concerns are the following:
Line 9: in the abstract, there are some placeholders that should be removed (introduction, materials and methods, results, discussion, conclusion).
Section 2.1: Details about the population's average age, body weight, height, and BMI are missing. Please provide them here divided per gender. If there is a gender imbalance (e.g. 15 males, 5 females) the findings of this study will not be considered relevant enough. In addition, I wish to stress that 20 subjects is not enough to provide relevant findings in general, even in the case of gender-balanced datasets. A suggestion in this case is to stress that this work is preliminary and be honest about the evident limitation of the study regarding the number of subjects and gender imbalances.
Section 2.3: detailed information about the informed consent form should be moved in section 2.1 instead of 2.3
Section 2.5: I spotted inconsistency with the nomenclature of movements. You referred to them as M1-M8, here you refer to them as Movement 1 - Movement 8. Please keep consistency throughout the whole paper.
Figure 10 is copyright-protected since it belongs to MediaPipe documentation. Please change it with one of your own.
Figure 11-12-13 should be bigger; they are impossible to read (individual images too small).
Section 3.4: I wonder about the relevance of decimals in joint angles (line 380). Is a difference of 0.15 actually visible, or is it just what the algorithm outputs? I wonder the same in the case of mV: are 4 relevant digits (the decimals) actually necessary? If we are considering mV, the fourth decimal is actually a 10-4 mV (hence, a 10^-7 V)
Section 3.5: Decimals in percentages are generally meaningless; I suggest showing the rounded values instead (97.48% becomes 98%).
Table 4: Accuracy should be rounded to 96%. Convert data of precision, recall, and f1-score to percentages as well. No information about what the "support" column is; please clarify in the text or eventually remove this column.
Discussion: Although I understand the validity of this study, I fail to picture this system being operated by people in their homes by themselves, even if they are not elderly. The system is based on a camera and a wearable device connected to a computer and operated by software that is not exactly easy to understand. Moreover, the considerations about the tested subjects apply here: your study was conducted on just 20 people, not enough to provide findings that can be applied to all the population without fault. As a result, I think this article is more a preliminary study instead of a real advancement in the field. These considerations should be pointed out in the discussion as well as all the other limitations of the proposed system.
Moreover, comparisons of the approach with other state-of-the-art methods in the literature (or generally a system with a higher level of accuracy and robustness, for example, a commercial VICON system) are missing; thus, the approach is missing validation. I suggest reviewing the works of Lancini et al. and Crenna et al. for insights.
Round 2
Reviewer 3 Report
Comments and Suggestions for Authors
Authors complied to my comments and now the article is suitable for publication.